# Classical Hodgkin Lymphoma: From Past to Future—A Comprehensive Review of Pathophysiology and Therapeutic Advances

**DOI:** 10.3390/ijms241210095

**Published:** 2023-06-13

**Authors:** Faryal Munir, Viney Hardit, Irtiza N. Sheikh, Shaikha AlQahtani, Jiasen He, Branko Cuglievan, Chitra Hosing, Priti Tewari, Sajad Khazal

**Affiliations:** 1Department of Pediatrics, Pediatric Hematology/Oncology, The University of Texas MD Anderson Cancer Center, Houston, TX 77030, USA; fmunir@mdanderson.org (F.M.);; 2CARTOX Program, Department of Pediatrics, Pediatric Stem Cell Transplantation and Cellular Therapy, The University of Texas MD Anderson Cancer Center, Houston, TX 77030, USA; vhardit@mdanderson.org (V.H.);; 3Department of Pediatrics—Patient Care, The University of Texas MD Anderson Cancer Center, Houston, TX 77030, USA; 4Department of Stem Cell Transplantation, The University of Texas MD Anderson Cancer Center, Houston, TX 77030, USA

**Keywords:** Hodgkin lymphoma, new drugs, brentuximab, checkpoint inhibitors, chimeric antigen T cells, targeted therapy

## Abstract

Hodgkin lymphoma, a hematological malignancy of lymphoid origin that typically arises from germinal-center B cells, has an excellent overall prognosis. However, the treatment of patients who relapse or develop resistant disease still poses a substantial clinical and research challenge, even though current risk-adapted and response-based treatment techniques produce overall survival rates of over 95%. The appearance of late malignancies after the successful cure of primary or relapsed disease continues to be a major concern, mostly because of high survival rates. Particularly in pediatric HL patients, the chance of developing secondary leukemia is manifold compared to that in the general pediatric population, and the prognosis for patients with secondary leukemia is much worse than that for patients with other hematological malignancies. Therefore, it is crucial to develop clinically useful biomarkers to stratify patients according to their risk of late malignancies and determine which require intense treatment regimens to maintain the ideal balance between maximizing survival rates and avoiding late consequences. In this article, we review HL’s epidemiology, risk factors, staging, molecular and genetic biomarkers, and treatments for children and adults, as well as treatment-related adverse events and the late development of secondary malignancies in patients with the disease.

## 1. Introduction and Historical Background

Hodgkin lymphoma (HL) is one of the most curable cancers in both pediatric and adult patients. The primary disease carries an excellent prognosis with an estimated 5-year survival rate of more than 98% [1]; however, long-term overall survival remains poor because of relapsed or refractory (R/R) disease and the late effects of treatment regimens. The current American Cancer Society statistics report that the 5-year relative survival rate for all patients diagnosed with HL is now about 87% [2].

HL is named after the English physician Thomas Hodgkin, who first described the disorder in 1832 [3]. In his publication, he noted that the unknown ailment was characterized by painless lymph node enlargement [3]. More than two decades later, in 1856, Samuel Wilks noted that splenomegaly was also a common symptom in patients with the disorder [4]. Finally, at the turn of the century, two pathologists (Carl Sternberg and Dorothy Reed) working independently of each other described the distinctive, multinucleated cells which now bear their names (Reed–Sternberg [RS] cells) and are pathognomonic for classical HL (cHL) [5]. Their discovery helped dispel the notion—commonly held within the medical community at the time—that Hodgkin’s disease was merely a form of tuberculosis (because both disorders commonly present with night sweats, weight loss, fever, and lymphadenopathy) [6].

HL is a rare malignancy of lymphoid origin that accounts for about 15% of all lymphoma diagnoses. The disease is characterized by unique, mononucleated Hodgkin cells and giant, multinucleated RS cells—collectively known as Hodgkin–Reed–Sternberg (HRS) cells—surrounded by inflammation [7,8]. The classical presentation involves supra-diaphragmatic lymphadenopathy, often associated with systemic (constitutional) B symptoms comprised of unexplained, high-grade fever; drenching night sweats; weight loss of at least 10% of body weight; and fatigue [9].

HL is sometimes incorrectly classified as being primarily a malignancy of adulthood. This is likely partially due to its bimodal age distribution, which consists of an incidence peak in young adults (ages 20–34 years) and a second peak in older adults (older than 55 years). The average age at diagnosis is 39 years [10]; however, HL is actually relatively common in the pediatric population as well [11]. In fact, HL accounts for approximately 7% of childhood cancers overall and 1% of childhood cancer deaths in the United States [10]. Additionally, it is the most common childhood cancer in the 15-to-19-year-old age group [12].

The World Health Organization broadly categorizes HL into two classes based on histopathological differences [13]: classical Hodgkin lymphoma (cHL) and nodular lymphocyte-predominant HL (NLPHL). cHL, which has four subclasses (described later in the classification section), is much more common and accounts for approximately 95% of HL cases, whereas NHLPL accounts for only 5% [8,14].

The conventional treatment of HL, which has been developed over the past several decades, consists of stage-driven chemotherapy with or without radiation and achieves a cure rate of about 80%. Alternate high-dose chemotherapies and hematopoietic stem cell transplantation (SCT) are the only second-line therapies available for R/R cases [8,15,16,17].

Despite the significant cure rates achieved using traditional therapies, about 20% of patients with cHL experience R/R disease, which carries a dismal prognosis; only half of such patients are cured via intensive second-line therapies [7,11,18]. Moreover, the potential late effects of treatment continue to be concerning, warranting the need for improvement in the field [1,19].

In clinical studies for R/R HL, novel methods such as monoclonal antibodies (i.e., brentuximab vedotin [BV, an anti-CD30 monoclonal antibody], nivolumab, and pembrolizumab) and immunomodulatory medications have produced outstanding results and are now being incorporated into front-line treatment regimens [20,21].

This manuscript provides a thorough overview of HL’s epidemiology, risk factors, staging and etiology, pathophysiology, molecular biology, clinical presentation, differential diagnosis, diagnostic workup, treatment, and treatment-related adverse events, as well as the late development of secondary malignancies and current and upcoming trends and viewpoints.

## 2. Materials and Methods

A comprehensive literature search was conducted to identify relevant studies and reviews pertaining to completed and ongoing clinical trials, including those by the Children’s Oncology Group (COG). The search was conducted on various databases, including PubMed, Google Scholar, and Clinicaltrials.gov (accessed on 8 May 2023), with a focus on studies investigating the efficacy of novel therapies for HL. The search was supplemented by a review of older references, where applicable, to ensure a comprehensive assessment of the available evidence. The search terms employed included “Hodgkin”, “lymphoma”, “new drugs”, “brentuximab”, “checkpoint inhibitors”, “chimeric antigen T cells”, and “targeted therapy”. In addition, abstracts from relevant conferences, such as the “American Society of Hematology” and “American Society of Clinical Oncology”, were also reviewed. To ensure completeness, the references from the articles and abstracts that were evaluated were cross-referenced.

## 3. Epidemiology

Every year, HL accounts for approximately 10% of newly diagnosed lymphoma cases (2.6 cases per 100,000 population) in the United States, or 2% to 2.5% of total cancer diagnoses. The American Cancer Society estimated that, in the United States in 2022, 8540 of the 89,010 lymphoma cases were HL; that HL would cause about 920 deaths (0.3% of all cancer deaths) [2]; and that HL would represent approximately 7% of childhood cancers overall and 1% of childhood cancer deaths [2]. According to the National Cancer Institute, from 2010 to 2020, the age-adjusted rate of new HL cases fell an average of 1.6% per year, while the age-adjusted death rate fell an average of 4.0% per year [22]. Whereas the incidence of most lymphomas increases with advancing age, HL is distinguishable because of its bimodal age distribution [22]. HL occurs most frequently in adolescents and young adults (aged 15–35 years) and in older adults (over the age of 55 years); the average age at diagnosis is 39 years [22]. HL is exceptionally rare in infants, toddlers, and pre-pubertal patients [22]. It remains the most common cancer diagnosed in adolescents aged 15 to 19 years [2].

With regard to race, HL incidence rates vary by region across the United States [23]. However, in general, non-Hispanic whites have slightly higher rates of the disease and American Indians/Alaska Natives and Asians/Pacific Islanders have lower rates [23,24]. In terms of sex predilection, in adults, HL is only slightly more common in men than in women; however, in pediatric patients, the difference between the sexes is significantly higher, and approximately 85% of patients are boys [25].

## 4. Risk Factors and Etiology

There is no clearly defined etiology for HL, but there are numerous risk factors that may predispose an individual to develop the malignancy [26]. For instance, individuals with immunodeficiencies, such as acquired immunodeficiency syndrome (AIDS), and those on chronic immunosuppression because of solid-organ or stem-cell transplants are at higher risk of developing HL [26,27] (although individuals with immunodeficiencies actually have a greater risk of developing non-HL than HL, and HL is not considered an AIDS-defining malignancy) [28,29]. 

Epstein–Barr virus (EBV) positivity has also been identified as a causative factor in HL [26,30]. In fact, EBV genetic material has been detected in the RS cells of some HL patients [30]. EBV affects 90% to 95% of adults worldwide and is associated with approximately 1% of all cancers and one-third of all HL cases [31,32]. However, only a small percentage of individuals infected with EBV actually develop lymphoma [32]. Therefore, it is likely that other biological or epidemiological determinants besides the virus itself play a role in the development of this disease [32,33]. Family history also appears to be a determinant in the development of HL [34]. Multiple population-based cohort studies in European countries have demonstrated that a family history of HL is an independent risk factor for the development of childhood HL [35]. One study found that the overall lifetime cumulative risk of HL in the first-degree relatives of a patient with HL was three times higher than that in the general population [36].

## 5. Pathophysiology and Molecular Biology of HL

HL is a complex, multifactorial malignancy that primarily involves B cells (although 1–2% of cases involve T cells) [37]. These lymphocytes originate in the germinal center of lymph nodes [38]. The disease is characterized by the presence of unique, mononucleated Hodgkin cells, and large, multinucleated RS cells, collectively known as HRS cells, and are rare amidst the extensive and complex inflammatory background. Though both Hodgkin and RS cells are abnormal lymphocytes, their morphological appearance differs, and both are highly specific to HL. RS cells continually develop from the Hodgkin cells as a result of incomplete cytokinesis and re-fusion [39]. Nonetheless, HRS cells are the hallmark of cHL, and are characteristically unable to express B-cell-specific genes, most notably the immunoglobulin (Ig) heavy-chain gene, and are thus unable to produce antibodies [37]. Various mechanisms (such as EBV infection) activate the anti-apoptotic nuclear factor kappa B (NF-κB) transcription factor signaling pathway [40]. Activation of this pathway prevents the apoptosis of the defective lymphocytes and promotes the proliferation of RS cells [37,40].

HRS cells of cHL exhibit a perplexing co-expression of markers from different hematopoietic cell types, including expression of the B-cell transcription factor PAX5 [39,41]. Genetic analysis studies confirmed that HRS cells are transformed B cells, as they possess Ig heavy- and light-chain V gene rearrangements specific to B cells, along with somatic mutations associated with germinal center (GC)-experienced B cells [39,42,43]. Furthermore, the detection of destructive mutations in IgV genes indicated the derivation of HRS cells from pre-apoptotic GC B cells [43]. Notably, a small subset of cHL cases may be attributed to T-cell origin [44]. HRS cells express the CD30 marker and show key features similar to normal CD30+ B cells, including mutated IgV genes, class switching, and expression of MYC [45]. Although the transformation process of pre-apoptotic GC B cells into HRS cells remains poorly understood, evasion of programmed cell death appears to be a crucial early event [39]. Interestingly, cases with mutations impairing BCR expression are often associated with EBV infection. Furthermore, EBV-infected HRS cells exhibit lower mutation loads, suggesting that viral gene expression substitutes for oncogene and tumor suppressor gene mutations, supporting the pathogenic role of EBV in EBV+ cHL [46]. The downregulation of the B-cell program in HRS cells involves multiple factors, including the dysregulation of transcription factors, immune evasion, and epigenetic silencing [39]. Notably, cHL lacks a defining genetic lesion, and the combination of genetic alterations may contribute to the uniqueness of this disease. HRS cells rely on aberrant, constitutive activity in several signaling pathways, including NF-κB, JAK/STAT, and PI3K/AKT, with the high constitutive activity of the NF-κB pathway being critical for HRS cell survival [39,47,48]. The most frequent mutations found in cHL cases are those involving regulators of these signaling pathways, including NF-κB factor REL, TNFAIP3, SOCS1, and STAT6 [39,48]. Multiple receptors, such as CD30 and CD40, transmit pro-survival and pro-proliferative signals via NF-κB, highlighting their potential involvement in HRS cell signaling [39].

## 6. Classification of HL

The World Health Organization classifies HL into two main categories, cHL and NLPHL, on the basis of morphological and immunohistochemical characteristics [13,49].

### 6.1. cHL

cHL is the larger and more common category (90–95% of cases of HL) and usually presents as an acute, aggressive illness. The presence of malignant HRS cells is the hallmark of cHL [8,14]; these cells are persistently found in all subtypes and are positive for cluster of differentiation (CD)15 in 85% of cases, positive for CD30 in 100% of cases, and are essentially negative for CD45 by immunohistology staining [7]. cHL is further classified into four histologic subgroups based on the structure, cell surface markers, percentage or density of Hodgkin and RS cells, and the amount of background inflammatory infiltrate [8]. The subtypes include nodular sclerosis HL, mixed cellularity HL, lymphocyte-rich HL, and lymphocyte-depleted HL [13].

Nodular sclerosis cHL is the most common subtype and carries a better prognosis than the other subtypes [13]. About 70% of cHL cases worldwide belong to this category. Nodular sclerosis cHL is characterized by neoplastic, lacunar-shaped RS cells with an enriched inflammatory background of sclerosing bands [50]. Additional features of the subtype are mediastinal lymphadenopathy in 80% of cases and large nodes of more than 10 cm in diameter (bulky disease) in about 50% of cases [51].

Mixed cellularity cHL (MCcHL) is more common in developing countries than in developed countries and is frequently associated with human immunodeficiency virus (HIV) infection. In the United States, it accounts for about 25% of cHL cases. MCcHL is characterized by a more diffuse, mixed inflammatory background without any sclerosis and with scattered HRS cells. It is also strongly associated with Epstein–Barr virus infection and has a relatively worse prognosis [52]. 

Lymphocyte-rich cHL may have a nodular or diffuse cellular background of small lymphocytes, without neutrophils or eosinophils. It accounts for about 5% of all cHL cases, usually presents early, is devoid of bulky disease, and carries an excellent prognosis with the current therapies [53,54,55]. 

Lymphocyte-depleted cHL is a rare subtype (<1% of cases) and is usually very aggressive with a poor prognosis. It shows diffuse infiltration of HRS cells without any inflammatory background [50,55].

### 6.2. NLPHL

NLPHL accounts for about 5% to 10% of HL cases, is indolent in most instances, and is considered to be a distinct disease entity that shares more characteristics with B-cell non-HL than with cHL [7,14,56]. The malignant cells in NPLHL—lymphocyte-predominant cells (popcorn cells)—are positive for CD20 and lack CD30. In contrast, RS cells are consistently positive for CD30 [50,56,57].

NLPHL is approached in a different manner both diagnostically as well as therapeutically, the discussion of which is beyond the scope of this article. In this review, we will focus on cHL.

## 7. Clinical Presentation and Symptomatology

Painless lymphadenopathy is the most common initial presenting sign of HL and is observed in approximately 80% of pediatric patients [58]. Lymphadenopathy associated with Hodgkin is typically firm and rubbery in consistency and is most often noted in the cervical, supraclavicular, and/or axillary regions. It is occasionally present in the inguinal region as well [59,60]. 

In addition to lymphadenopathy, nearly 75% of children diagnosed with HL will present with a mediastinal mass [61]. Mediastinal masses may present at any age, but they are more common in children older than 12 years of age [62]. Patients with mediastinal masses may have associated dyspnea, orthopnea, or dysphagia [62]. These patients are at risk of rapid respiratory compromise and/or decompensation, particularly if they require sedation, and also are at risk of superior vena cava syndrome [63].

B symptoms are another common presenting sign in patients with HL [20,64]. These consist of fatigue, fever, drenching night sweats, anorexia, and/or weight loss (≥10% loss within six months preceding diagnosis) [20]. Although these symptoms are commonly associated with HL, they are not pathognomonic for the malignancy, as they can present in patients with a multitude of other illnesses (including non-HL) [20]. Nevertheless, the presence or absence of B symptoms is sufficiently important to be incorporated into the staging system for HL (see the staging subsection below) [65,66].

## 8. Differential Diagnoses

The differential diagnoses for HL may be divided into three primary categories: infections, inflammatory or autoimmune disorders, and other malignancies [59]. Although many infectious processes can present with the same painless lymphadenopathy and fevers often noted in patients with HL, the most critical infections to exclude are HIV, tuberculosis, and infectious mononucleosis (especially given the close association between HL and EBV) [59]. With regard to inflammatory or autoimmune disorders, conditions such as systemic lupus erythematous, rheumatoid arthritis, sarcoidosis, and hemophagocytic lymphohistiocytosis should always be considered as a differential diagnosis for a patient suspected of having HL. Other malignancies—particularly other lymphomas, such as non-HL, diffuse large B-cell lymphoma, and anaplastic large-cell lymphoma—must be considered as well [59,67].

## 9. Diagnostic Workup

Because of its varied clinical presentation and similarity to an array of other conditions, HL is sometimes initially misdiagnosed, which may result in critical delays in the initiation of therapy and a poor prognosis. For this reason, a meticulous diagnostic workup is critical for any patient suspected of having HL. Table 1 summarizes the essential components of the diagnostic workup for HL.

## 10. Staging and Risk Stratification

HL treatment is guided by the risk category of the diagnosis, which is based on the disease stage and clinical presentation [73]. Physical examination and radiological imaging are used to stage the disease [58] and to acquire information regarding several factors of varying prognostic significance [74]. These factors include the location and laterality of the affected lymph nodes on either or both sides of the diaphragm, the number of lymph nodes involved, the involvement of contiguous lymph nodes, and the presence of bulky disease (defined as a mediastinal mass >1/3 the size of the intrathoracic diameter on a posterior–anterior chest X-ray or any mass ≥10 cm on a computed tomography [CT] scan at diagnosis [74]). Of note, the Children’s Oncology Group has a lower threshold for bulky disease and defines it as a mass greater than or equal to 6 cm in pediatric populations in the presence or absence of B symptoms [75].

There are two main classification systems for staging HL. The Ann Arbor Classification System, developed in 1971 by the Committee on Hodgkin’s Disease Staging Classification, was the first widely used staging system for HL [76]. The original Ann Arbor system was primarily based on anatomy and was similar to the Tumor, Node, Metastasis staging system for solid tumors devised by Pierre Denoix between 1943 and 1952. In 1988, the Cotswolds-modified Ann Arbor classification system added substaging variables (A, B, E, and X) to the original Ann Arbor system (Table 2) [77]. The Cotswolds-modified Ann Arbor classification system is the current standard and remains the most commonly used classification system for pediatric and adult HL [57,58,66,76].

HL has four clinical stages. Stages I and II indicate limited/early disease, while stages III and IV indicate advanced disease. In pediatric HL cases, 19% are stage I, 49% are stage II, 19% are stage III, and 13% are stage IV at diagnosis [78].

Clinical disease risk stratification is performed based on disease stage, bulk status, and presence or absence of B symptoms. Three risk groups are determined according to the Children’s Oncology Group (COG) classification [73]: Low-risk group: stage IA/IIA without bulk or extra-nodal extension (E).Intermediate-risk group: stage IIB/IIIA, as well as stage IA/IIA with extra-nodal extension (E) or bulky disease.High-risk group: includes all stage IIIB and IV patients [7,73].

In addition, in 2011, the Lugano Classification system (Table 3) was devised at the International Conference on Malignant Lymphoma [79]. This new staging system sought to modernize the Cotswolds–Ann Arbor system [79]. One of the most impactful innovations of the Lugano system was the integration of positron emission tomography (PET)/CT scanning in the initial evaluation of patients with HL [79]. The incorporation of PET/CT scanning as a baseline assessment tool eliminated the need for bone marrow aspiration/biopsy and improved the accuracy of disease staging accuracy, resulting in fewer patients being undertreated or overtreated [79]. Additionally, the Lugano system eliminated the “X” staging suffix of the Cotswolds–Ann Arbor system that had previously denoted the presence of bulky disease. The Lugano system continued to use the “A” and “B” staging suffixes to signify the presence or absence of B symptoms [79].

## 11. Treatment Strategies for Classical HL

The most popular treatment strategies for cHL are combined chemotherapy, radiation, and—more recently—immunotherapy. Despite recent advancements in immunotherapy, the first-line treatments for HL continue to evolve [80]. Radiation has been a cornerstone of HL therapy since the early 1900s [80]. However, because of the high survivability of the disease (the 5-year relative survival for patients diagnosed with cHL at age 0–19 years is higher than 95% [81]) and the high rate of long-term side effects, efforts have been made in recent years to reduce the usage of radiation in pediatric populations [75].

### 11.1. Combination Chemotherapy for HL

Combined chemotherapy (i.e., chemotherapy with or without radiation therapy) remains the most commonly used treatment modality for HL patients across all age groups [80]. Because of ongoing research and emerging molecular and cellular therapies, the treatment approaches have evolved over the past several decades [80] and are slightly different in adult and pediatric patients.

### 11.2. Treatment of Adult cHL

For adults with cHL, treatment is mainly guided by disease staging.

#### 11.2.1. Stage I–II/Early Stage HL

For a long time, radiotherapy alone was the standard treatment option for adult early stage HL [82]. Later on, chemotherapy was incorporated into treatment regimens once it was established that it was safe and efficacious and that, when it was given in combination with radiotherapy, the risks to the patient were less than those associated with the use of radiotherapy alone (e.g., cardiotoxicity, pulmonary disease, and secondary malignancies, the most dreaded long-term side effects) [83]. Four cycles of doxorubicin (adriamycin), bleomycin, vinblastine, and dacarbazine (ABVD) followed by involved-field radiotherapy (IFRT) at a dose of 36 Gy was the first recognized standard-of-care treatment for adults with early stage HL [84]. Studies later established that a slightly reduced IFRT dose of 20 Gy was equally effective [85].

The German Hodgkin Study Group (GHSG) study trials HD10, HD13, and HD16 consistently established that two cycles of ABVD followed by consolidation radiotherapy/involved-site radiotherapy (ISRT) with 20 Gy was effective as a primary treatment approach for patients with early stage favorable disease. The HD10 trial reported an overall survival (OS) rate of 94% and a progression-free survival (PFS) rate of 87%. In addition, it showed no significant differences between patients who received four vs. two cycles of ABVD in terms of the 5-year OS (97.1% vs. 96.6%) [17,85,86,87].

The UK RAPID trial looked at the possibility of avoiding radiotherapy altogether. Patients with a negative interim PET scan (Deauville score = 1 to 2) were randomized to receive IFRT vs. observation only following three cycles of ABVD. The results showed a marked difference in the 3-year PFS rates of patients who received combined modality therapy vs. those who received chemotherapy alone (97.1% vs. 90.8%) [88]. Similar results were seen in the GHSG HD16 trial (5-year PFS rates: 93.4% for ABVD plus consolidative radiotherapy vs. 86.1% for ABVD alone) [89] and in the European Organization for Research and Treatment of Cancer H10 trial (5-year PFS rates: 99.0% for ABVD plus radiotherapy vs. 87.1% for ABVD only) [86].

Subsequently, the H10U trial suggested the role of interim PET scanning in facilitating early treatment adaptation. Patients with stage I or II disease (favorable or unfavorable disease) who had a positive PET scan after two cycles of ABVD underwent treatment escalation with two cycles of bleomycin, etoposide, doxorubicin, cyclophosphamide, vincristine, procarbazine, and prednisone (BEACOPP) plus INRT, which improved the 5-year PFS to 90.6% versus 77.4% when receiving two additional cycles of ABVD and INRT only [86].

As of 2020, the National Comprehensive Cancer Network (NCCN) recommends treating patients with favorable stage IA to IIA disease with a combined modality approach consisting of two cycles of ABVD followed by a reassessment with an interim PET scan. Patients with a PET Deauville score of 1 to 3 should be treated with an ISRT course of 20 Gy. For patients with a PET Deauville score of 4, two additional cycles of ABVD followed by interim PET/CT may be considered prior to ISRT (30 Gy) versus two cycles of escalated BEACOPP, also known as eBEACOPP (bleomycin, etoposide, doxorubicin, cyclophosphamide, vincristine, procarbazine, and prednisone), followed by restaging with PET before proceeding to ISRT is recommended. All patients with a PET Deauville score of 5 should undergo biopsy at the end of treatment [9]. For additional recommendations, please refer to jccn.org (accessed on 21 January 2023) [9].

The treatment for early stage unfavorable cHL in adults differs slightly from that for children; most groups suggest treating patients with four cycles of ABVD (over BEACOPP with ABVD) followed by 30 Gy radiotherapy as the preferred choice of therapy [90,91,92,93]. In the GHSG’s HD14 trial for stage I-II unfavorable disease, much higher toxicity (56–87%) was observed in the BEACOPP/ABVD arm [94].

#### 11.2.2. Stage III–IV/Advanced HL

Advanced HL is usually treated with longer-duration chemotherapy alone (i.e., more cycles than are used for early stage disease [8]). The current international standard of care is to treat adult patients with advanced HL with six cycles of ABVD. Additional PET-guided radiotherapy (30 Gy) is reserved for patients with initial bulky or residual disease [92,95].

The final report of the Italian randomized control trial GITIL/FIL HD0607 (NCT00795613) concluded that, regardless of the lymph nodal mass size at presentation, consolidative radiotherapy can be safely avoided in patients with advanced-stage HL who show a complete metabolic response with negative PET-2 and PET-6 scan with ABVD. In addition, this trial examined the benefits of intensified treatment with eBEACOPP for patients with a positive interim PET scan (Deauville 4–5) and reported a 3-year PFS of 60% and an OS of 89% [96]. The intergroup trial S0186 showed similar results with a 3-year PFS of 65% and an OS of 97% for patients who had a positive interim PET scan [97,98]. 

The Response-Adapted Therapy in Advanced Hodgkin Lymphoma (RATHL) trial evaluated the role of interval PET-scan-guided treatment for advanced HL disease. Patients received two cycles of ABVD followed by PET scanning. Patients with a PET Deauville score of 1 to 3 received four more cycles of either ABVD or doxorubicin, vinblastine, and dacarbazine (AVD). There were no significant differences in the 3-year PFS and OS rates between the two groups, except that removing bleomycin from the ABVD regimen markedly decreased the incidence of pulmonary toxicity in the patients who received AVD. In contrast, patients with a PET Deauville score of 4 to 5 (i.e., those with positive interim PET scans) were given eBEACOPP treatment and had 3-year PFS and OS rates of 71% and 85%, respectively [99]. However, this trial did not investigate the role of radiotherapy. 

The European Organization for Research and Treatment of Cancer (EORTC), the Lymphoma Study Association (LYSA), and the GHSG have established that six cycles of eBEACOPP followed by PET-guided radiotherapy to PET-positive sites (>2.5 cm) is the standard of care for advanced-stage HL on the basis of several studies demonstrating the persistent efficacy of eBEACOPP. Among these studies is the HD15 trial, which included stage III to IV and IIB patients with large mediastinal nodes or disease involving extranodal sites [16,100,101]. In a meta-analysis of 14 different studies, patients with advanced-stage HL who were treated with eBEACOPP had a statistically significant survival benefit of 7% compared to those treated with ABVD [102]. 

To reduce the toxicity associated with eBEACOPP, subsequent studies focused on de-escalating strategies [8]. The GHSG HD18 study demonstrated no significant difference in PFS following four cycles of eBEACOPP as compared with six or eight cycles (92.2% vs. 90.8%) when investigated on an interim, PET-adapted design monitoring PET scan responses after two cycles of eBEACOPP [103]. This led to the conclusion that four cycles of eBEACOPP alone can be sufficient to treat most patients with advanced cHL while limiting toxicity [8]. 

Subsequently, the LYSA AHL2011 trial concluded that, after two initial cycles of eBEACOPP, PET-guided de-escalation to ABVD is plausible for patients with newly diagnosed, advanced-stage HL (stage IIB bulk or stage III to IV) and PET Deauville scores of 1 to 3 [8,104]. Treatment-related toxicities were also markedly less frequent in the PET-adapted ABVD treatment group [104]. 

BV combined with a modified eBEACOPP regimen (BrECADD) is currently being evaluated in the ongoing GHSG HD21 trial as a different strategy to further reduce chemotherapy-associated acute and long-term toxicity in patients with advanced-stage cHL [105].

The ECHELON-1 randomized control trial investigated the effectiveness of the BV-AVD vs. the ABVD regimen [106]. Patients in each group received six cycles of therapy, and no modifications were made on the basis of interim scans. Although the PFS rate was improved in the BV-AVD cohort compared to the ABVD cohort (84.3% vs. 73.7%) because of the removal of bleomycin, and pulmonary toxicity was also lower, more hematologic toxicity and an increased incidence of peripheral neuropathy were, however, observed in the BV-AVD cohort [107].

According to the 2020 NCCN guidelines [9], ABVD is recommended as the preferred regimen, to be given for two cycles, and afterwards PET scanning is used for restaging (as established by the RATHL trial); then, patients with a PET Deauville score of 1 to 3 receive four cycles of AVD treatment. Following four cycles of AVD, the therapeutic options for initially bulky or specifically chosen PET-positive areas include observation or ISRT. Patients with a PET Deauville score of 4 can either receive two more cycles of ABVD (for a total of four cycles) or two cycles of eBEACOPP, followed by a PET scan to determine the patient’s response. For patients with a PET Deauville score of 5, a biopsy is advised; however, in some circumstances, two cycles of BEACOPP may be given. For patients with a PET Deauville score of 4, treatment continues as described earlier if a biopsy is negative. BV plus AVD is initially given for two cycles in patients with stage III or IV HL, followed by restaging with PET scanning. A total of four further cycles of BV plus AVD are administered to patients with a PET Deauville score of 1 to 4. If a patient has a PET Deauville score of 5, alternative therapy for refractory HL should be investigated if the biopsy results are good. Patients may be monitored or given ISRT to PET-positive areas if end-of-therapy PET scanning yields a PET Deauville score of 3 or 4. For additional recommendations, please refer to jccn.org (accessed on 21 January 2023) [9].

### 11.3. Treatment of Pediatric cHL

An essential management concept for pediatric cHL in both North America and Europe is to base the intensity of therapy on the disease’s risk stratification [7,73]. Risk categories are defined earlier in the staging section; the tumor stage, the severity of the disease, and the existence of B symptoms are the key criteria used to classify patients’ risks. The use of radiotherapy in ongoing COG and European Network Pediatric Hodgkin Lymphoma (EuroNet-PHL) trials is heavily influenced by patients’ responses to chemotherapy (typically measured with PET scans). This approach aims to reduce the long-term adverse effects associated with the use of radiotherapy in pediatric populations [7]. To improve outcomes in higher-risk patients, both study groups have switched to lower radiotherapy volumes that target the highest-risk locations using highly conformal methods; these lower radiotherapy volumes are combined with normal or intensified chemotherapy regimens [7,57]. Immunoregulatory medications are currently being studied in ongoing trials due to the potential acute toxicity of increased chemotherapy [73,108].

#### 11.3.1. Low-Risk cHL

Pediatric low-risk cHL carries an excellent prognosis overall. For low-risk cHL (stage IA and IIA without bulk or extranodal extension), there is no unanimity in the COG regarding the optimal therapeutic regimen [73]. The COG used the default ABVD background (doxorubicin/bleomycin/vinblastine/dacarbazine) to create modified regimens aimed at reducing the toxicity of drugs, especially dacarbazine and vinblastine (which were replaced with etoposide and vincristine, respectively). In the POG9226 trial, a 6-year event-free survival (EFS) rate of 91% was reported with four cycles of doxorubicin, bleomycin, vincristine, and etoposide (ABVE) followed by IFRT [109]. 

Some other regimens investigated in different studies for low-risk cHL include vinblastine, doxorubicin, methotrexate, and prednisone (VAMP; four cycles) [110] and cyclophosphamide, vincristine, procarbazine, and prednisone (COPP; four cycles) with low-dose IFRT; results have been adequate [111]. Later trials looked at the possibility of reducing the number of chemotherapy cycles [112].

Subsequently, the COG AHOD0431 trial investigated a lower-intensity regimen—doxorubicin, vincristine, prednisone, and cyclophosphamide (AVPC)—in children and adolescents with low-risk cHL. All patients received three cycles of chemotherapy followed by a response-based approach to decide about further therapy. No further treatment was given to patients who had a complete response, defined as a nodal mass reduction of greater than 80%, a return to normal size node/organ on a CT scan, or a negative PET/gallium scan. Patients with a partial response after three cycles received 21 Gy of IFRT. Patients who developed a low-risk relapse (as defined by the protocol) after chemotherapy qualified for a salvage regimen with vinorelbine, ifosfamide, dexamethasone, etoposide, cisplatin, and cytarabine (DECA) with 21 Gy of IFRT [73,113]. Most (88%) patients achieved a complete response without needing high-dose therapy, autologous stem cell rescue, or IFRT, and the OS rate was 99.6%. However, the study was closed early because of a higher-than-expected relapse rate and a drop in EFS below the goal. A secondary analysis of the data revealed that EFS was actually significantly better in patients with MCcHL vs. those with nodular sclerosis (97.2% vs. 75.8%). This finding suggested that initial treatment with AVPC is a reasonable choice for patients with mixed cellularity histology, but that patients with nodular sclerosis require additional or alternate treatment strategies. Additionally, the researchers discovered that a decreased erythrocyte sedimentation rate of less than or equal to 20 mm/hour and a negative post-cycle 1 PET scan are associated with improved EFS [113]. 

The GPOH-HD-95 and GPOH-HD-2002 trials also consistently showed that radiotherapy can be excluded in TG-1 patients who achieve CR after chemotherapy [19,114,115].

#### 11.3.2. Intermediate-Risk cHL

The pediatric intermediate-risk HL population was investigated in the COG phase 3 trial AHOD0031, which evaluated early chemotherapy response assessments as a means of dictating subsequent therapy [75]. Two cycles of ABVE with prednisone and cyclophosphamide (ABVE-PC) were administered to patients with newly discovered, intermediate-risk HL; the patients then received an early response assessment (ERA) using a PET or CT scan. Patients who showed an adequate response received two more cycles of ABVE-PC, followed by a response evaluation. Rapid early responders (RERs) with complete responses were randomly assigned to receive IFRT vs. observation only, while all RERs with partial responses nonrandomly received IFRT (21 Gy). Patients who were slow early responders (SERs) after two cycles of ABVE-PC were randomly assigned to receive or not receive two cycles of DECA-based chemointensification before receiving two more cycles of ABVE-PC. In addition, all SERs received IFRT. The 4-year OS and EFS rates in this study were 97.8% and 85%, respectively. There was no discernible difference in the 4-year EFS rate between patients who received IFRT against those who did not (87.9% vs. 84.3%, respectively) among RER patients who had CR at the conclusion of treatment. In addition, there were no significant differences in 4-year EFS rates in the patients who received DECA compared with those who did not. Overall, this trial concluded that additional chemotherapy can be supplemented in SERs with positive PET scans and that radiotherapy can be omitted in RERs with a complete response at the end of chemotherapy [75,116]. 

The Childhood Hodgkin International Prognostic Score (CHIPS) was created using four parameters from a predictive model for EFS (stage IV illness, large mediastinal adenopathy, low serum albumin, and fever). Having no more than one CHIPS risk factor (CHIPS score = 0–1) was associated with a 4-year EFS of around 90% for patients receiving four cycles of chemotherapy plus IFRT, whereas having two or more CHIPS risk factors (CHIPS score = 2–3) predicted a 4-year EFS of approximately 78% [116,117]. This finding indicated that this increment of therapy beyond four cycles of ABVE-PC should be considered for patients with more than one CHIPS risk factor [117].

#### 11.3.3. High-Risk cHL

The Children’s Cancer Group study 59704 assessed the effectiveness of treating pediatric high-risk cHL with upfront, intensified chemotherapy with BEACOPP [57,118]. Four cycles of BEACOPP were initially given to all patients. Further chemotherapy included four more cycles of BEACOPP for SERs (total = 8 cycles), two cycles of ABVD for male RERs, and two cycles of doxorubicin, bleomycin, and vinblastine (ABV)/COPP for female RERs; all patients then received IFRT. The study showed remarkable results; the 5-year EFS rate was 94%, and the OS rate was 97%. However, long-term toxicities from BEACOPP emerged as a concern [57,73,118].

The Pediatric Oncology Group study P9425 was the first to formulate the ABVE-PC regimen, which later became the spine of all following COG trials for pediatric cHL [119]. P9425 originally investigated the effectiveness of ABVE-PC in patients with intermediate- or high-risk cHL; all patients were given three cycles of ABVE-PC followed by an early response evaluation. RERs then received IFRT (21 Gy), and SERs received two additional cycles of ABVE-PC (total = 5 cycles) followed by IFRT. This study reported an average 5-year EFS rate of 84% among the intermediate- and high-risk cHL groups [119].

Later, the COG investigated a modified version of ABVE-PC in the phase 3 trial AHOD0831, with the aim of reducing radiation intensity and exposure to alkylating agents [120]. All patients received two cycles of ABVE-PC. Then, RERs received two additional cycles of ABVE-PC and IFRT (limited to the initial bulk site), and SERs received two cycles of ifosfamide and vinorelbine (IV) followed by two more cycles of ABVE-PC and then radiotherapy (limited to the initial bulk site or the slow-responding sites). The study showed a 91.9% four-year EFS, which was slightly less than the projected goal of ≥95% [120]. However, the overall data were comparable to that from other trials for high-risk HL, with 5-year first EFS and OS rates of 79.1% and 95%, respectively [73,118,119].

A recently completed phase 3 trial, COG AHOD1331 (NCT02166463), investigated the incorporation of the immunotherapy agent BV into the doxorubicin, bleomycin, vincristine, etoposide, prednisone, and cyclophosphamide (AVE-PC) chemotherapy regimen and compared BV-AVE-PC with the current standard of care, ABVE-PC [57,73,121,122]. The study showed exciting results, which were presented at the 2022 American Society of Clinical Oncology Annual Meeting [121]. Specifically, the study reported 3-year EFS and OS rates of 92.1% and 99.3%, respectively, in the BV-AVE-PC group as compared with 82.5% and 98.5%, respectively, for the ABVE-PC group. This trial essentially indicated that BV-AVE-PC should be the new standard frontline therapy for pediatric cHL [121]. 

A group of investigators at New York Medical College is currently using a risk-adapted therapy approach in a pilot, non-randomized trial (NCT02398240) evaluating the efficacy of upfront BV with combination chemotherapy for the treatment of children, adolescents, and young adults with newly diagnosed cHL [108]. Patients received doxorubicin, vinblastine, dacarbazine, BV, and rituximab for four or six cycles. After the first two cycles, an early response evaluation was performed. Only high-risk patients with bulky disease and/or a sluggish or partial response to chemo-immunotherapy were eligible for IFRT. The preliminary results from the intermediate- and high-risk cHL groups are astonishing, with 100% CR, EFS, and OS rates at a median follow-up of 5 years, in addition to demonstrating a decrease in the use of radiotherapy [108]. With these results, the trial evaluated BV in patients with newly diagnosed, low-risk cHL in an effort to make BV a frontline therapy across all stages of newly diagnosed cHL.

The NCCN currently recommends that patients with high-risk cHL enroll in an ongoing clinical trial. Alternatively, these patients can receive treatment per the AHOD1331 protocol described above or the EuroNet-PHL-C1 protocol described below [57].

#### 11.3.4. Overview of the EuroNet-PHL Studies for the Treatment of Pediatric cHL

With an aim of standardizing cHL treatment across all European countries, the EuroNet-PHL C1 and C2 studies established protocols using the ERA approach and chemotherapies rationalized to reduce long-term adverse effects, especially gonadal and cardiovascular toxicities [73,123]. The studies were developed using a similar approach to that of the GPOH-HD-2002 study, which adopted the vincristine, procarbazine, prednisone, and doxorubicin (OPPA) chemotherapy regimen by substituting etoposide for procarbazine (OEPA) for males during induction. Likewise, in the cyclophosphamide, vincristine, prednisone/prednisolone, and procarbazine (COPP) regimen, dacarbazine was substituted for procarbazine (COPDAC) for females during consolidation [57,114]. Additionally, the EuroNet-PHL studies strove to decrease exposure to other agents such as vincristine. PET-CT or PET-magnetic resonance imaging were used for ERA [73].

The EuroNet-PHL C1 study stratified patients into three treatment groups using the Ann Arbor stages (Table 4). Treatment group (TG) 1 included stages IA/B and IIA; TG-2 included stages IEA/B, IIEA, IIB, and IIIA; and TG-3 included stages IIEB, IIIEA/B, IIIB, and IVA/B [73,123].

In the EuroNet-PHL C1 study, all treatment groups initially received two cycles of OEPA, followed by an ERA. For TG-1, patients with an adequate response continued with follow-up only, i.e., no further treatment was given. Those with an inadequate response proceeded to radiotherapy. For TG-2 and TG-3, following two cycles of OEPA, patients were randomized to receive two cycles of COPP vs. four cycles of COPDAC. Patients with an inadequate response also received radiotherapy. The 4-year OS and EFS rates were 98% and 88%, respectively, according to a preliminary analytic report. Patients with or without radiotherapy had EFS rates of 88% and 87%, respectively [57,123]. Additionally, in TG-2 and TG-3, there were no differences in the EFS rates between the COPP and the COPDAC arms. An erythrocyte sedimentation rate over 30 and bulky illness were linked to a worse EFS rate in TG-1 [73,123]. This trial demonstrated that dacarbazine can replace procarbazine in COPP and that radiotherapy can be discontinued in patients with negative PET scans following treatment [57,123].

The EuroNet-PHL C2 trial defines three treatment levels. TL-1 contains former TG-1 without risk factors; TL-2 consists of former TG-2 and also includes previously TG-1 with risk factors; and TL-3 is similar to former TG-3 [73]. Like the patients in the EuroNet-PHL C1 trial, patients in the ongoing EuroNet-PHL C2 trial initially receive two cycles of OEPA [73]. Patients in TL-1 with an adequate response proceed to one cycle of COPDAC-28, whereas those with an inadequate response receive radiotherapy, while in TL-2 and TL-3, patients are randomized to receive the standard COPDAC-28 vs. intensified DECOPDAC-21 (doxorubicin, etoposide, cyclophosphamide, vincristine, prednisone/prednisolone, and dacarbazine). TL-2 patients receive only two cycles of the assigned arms, whereas TL-3 patients receive four. Radiotherapy is limited to only patients who have an inadequate response at the ERA. In addition to lowered radiotherapy target volumes, EuroNet-PHL C2 is assessing the use of protons. It may be possible to lessen the late effects by reducing the dose of radiation to the nearby normal tissues (organs at risk), as a result of the steep proton dose falloff [73].

### 11.4. Summary of Combination Chemotherapy Approaches

Table 5 summarizes the HL risk stratification and treatment regimens from numerous clinical trials. The risk stratification system seeks to avoid overtreating low-risk patients (thereby minimizing the risk of long-term adverse effects, such as organ toxicity and secondary malignancies) while also avoiding undertreating high-risk patients (which could potentially increase the risk of relapse) [7].

## 12. Treatment of R/R HL

Most patients with HL, particularly pediatric patients, are cured after undergoing treatment with first-line therapy options [129]. However, despite the astounding developments in the treatment of HL and the achievement of high cure rates, 10% to 25% of patients develop R/R disease. Patients with advanced-stage cHL are more likely to relapse or have refractory disease and are challenging to treat [7].

For those individuals for whom initial therapy fails, several novel therapeutic approaches are available, including stem cell transplantation, molecular therapies, chimeric antigen receptor T-cell therapy, and immunotherapy [129].

### Autologous Stem Cell Transplant in HL

The conventional treatment for chemosensitive, relapsed cHL is reinduction (also known as salvage chemotherapy), followed by high-dose chemotherapy and autologous stem cell transplantation (HDC/ASCT), albeit the results for high-risk cHL relapse patients are still not ideal [7,130]. Studies indicate that 50% to 65% of pediatric patients with R/R cHL achieve remission with this approach [7,131,132]. However, to date, there is no gold-standard therapy available for this group of patients [130]. Standard-dose chemotherapy and radiotherapy may be used to treat some low-risk patients with late relapses and limited stage [7]. Like adult patients, juvenile/pediatric patients have a dismal prognosis, which is correlated with a diagnosis-to-first-relapse interval of less than a year [7,133]. Additionally, patients who fail to respond to salvage regimens are deemed to have a worse outcome with HDC/ASCT in most cases [7]. Thus, allogeneic stem cell transplantation (allo-SCT) or novel molecular therapies are called in as alternative options for treatment [7].

Extranodal involvement, primary progression of the disease, poor chemosensitivity to salvage therapy, and a shorter time duration to relapse are some of the most common adverse prognostic variables [130,134,135,136,137,138]. In addition, poor performance scores, bulky disease, the presence of B symptoms at relapse, relapse within an area previously exposed to radiation, conditioning with two or more salvage regimens, and disease persistence in areas that were initially PET-positive are associated with considerably worse outcomes [130].

Several salvage regimens, such as BEAM [BCNU (carmustine), etoposide, cytarabine, melphalan]; low intensity or miniBEAM; MINE (mitoguazone, ifosfamide, vinorelbine, and etoposide); alternating IEP (ifosfamide, etoposide, and prednisolone) with the standard ABVD; and ICE (ifosfamide, carboplatin, and etoposide), have been developed over many years [132,137,139,140]. Salvage with ICE has shown to achieve a cumulative remission rate of 88% in R/R pediatric and adult patients, but at the cost of severe myelosuppression and secondary malignancy as a result of exposure to alkylating agents [7,140].

In the last decade, other vinorelbine-based salvage regimens have emerged. These include ifosfamide plus vinorelbine (IV), which was investigated in the COG trial AHOD00P1 [141], and gemcitabine plus vinorelbine (GV), which was investigated in another phase 2 study [142]. These combination approaches for R/R cHL patients showed improved response rates of about 76% [141,142].

Most recently, a group at the University of Texas MD Anderson Cancer Center studied post-HDC/ASCT prognoses for pediatric and adult patients with cHL during a period of 15 years (2005–2019). The group reported that the post-HDC/ASCT outcomes improved over this time period in this group [130]. The study compared the standard salvage regimen BEAM with BuMel (busulphan/melphalan), Gem (Gemcitacine)/BuMel, and vorinostat/Gem/BuMel. Five-year PFS and OS rates with vorinostat/GemBuMel were better than the other combination at 72% and 87%, respectively. Additionally, brentuximab exposure prior to transplant and salvage with vorinostat plus gemcitabine and BuMel was associated with improved outcomes as combined as well as independent factors [130].

According to a report published recently by the Center for International Blood and Marrow Transplant Research (CIBMTR), significant survival gains were seen over the last two decades in HL patients who have received ASCT. This could be attributable to the availability of newer therapies to treat R/R HL. The 3-year survival probabilities (95%CI) were 72% (71–74%), 82% (81–83%), 88% (87–89%), and 92% (91–93%) for the time periods 2001–2005, 2006–2010, 2011–2015, and 2016–2019, respectively (Figure 1) [143].

## 13. Relapse after ASCT

Despite the reasonable success rates of ASCT, a good percentage of patients still relapse. Major factors increasing the risk of disease relapse after ASCT include active disease at the time of transplant, in addition to the high-risk features described earlier (e.g., the duration of remission after initial treatment (if <12 months, the risk of relapse is higher), primary refractory disease, the presence of B symptoms, and extranodal or bulky disease) [130,134,135,136,137,138,144,145,146]. Post-transplant maintenance and bridging therapies have been developed in an effort to minimize the aforementioned unfortunate risk of disease recurrence [144].

The combination of cyclophosphamide, vincristine, procarbazine, and prednisone (C-MOPP) is commonly used as a bridging therapy, although other options are available as well [147]. After the transplant, many institutions incorporate BV for maintenance, as studies have shown that the drug increases the rate of PFS over observation alone [147]. A common brentuximab maintenance regimen is 1.8 mg/kg (to a maximum dose of 180 mg) every 3 weeks beginning 30 to 45 days after ASCT for up to 16 cycles [147].

## 14. Novel Therapies for R/R HL

GC B cells are the source of HRS cells [56,148]. HRS cells are surrounded by a heterogeneous inflammatory infiltrate made up of T cells, B cells, natural killer cells, and mast cells, which collectively comprise less than 5% of tumor cellularity [108,148]. Through the release of several cytokines, this tumor microenvironment (TME) sends multiple survival signals that aid HRS cells’ survival [148,149]. HRS cells exhibit high levels of CD30 cell surface markers [108,150]. Interestingly, it has been demonstrated that about half of the infiltrating cells in the TME are regulatory B cells [151]. Targeting suppressive CD20+ regulatory B cells in the cHL TME may be advantageous, even in patients whose HRS cells lack CD20 expression [108,151].

In addition, the distinct biology of cHL is inspiring the development of newer molecular therapies [152]. RS cells exhibit genetic amplification of the 9p24.1 locus; further analysis revealed their overexpression of programmed death ligand 1 (PD-L1) and programmed death ligand 2 [152,153]. This discovery led to the rational development of PD-1 checkpoint inhibitors. Later clinical trials showed high response rates with the use of these checkpoint inhibitors for the treatment of advanced HL [152,154,155]. Pembrolizumab and nivolumab are the two most widely used PD-1 checkpoint inhibitors, and are currently approved by the US Food and Drug Administration (FDA) as third-line therapies for R/R cHL [152].

### 14.1. CD30 Inhibitor (Brentuximab Vedotin)

The HRS cells show high levels of CD30 expression, making them attractive therapeutic targets [108,150]. BV is an anti-CD30 antibody–drug conjugate that preferentially attacks CD30+ malignant HRS cells, causing their apoptosis [108]. 

The randomized, controlled ECHELON-1 trial investigated the effectiveness of combination chemotherapy with BV-AVD against the standard ABVD regimen in patients at least 18 years old who had advanced-stage cHL [106]. The study showed that PFS was improved in the BV-AVD cohort compared with the ABVD cohort (84.3% vs. 73.7%), and a decreased rate of pulmonary toxicity was also observed in the BV-AVD cohort, likely due to removal of bleomycin. However, in the BV-AVD cohort, greater hematologic toxicity and an increased incidence of febrile neutropenia peripheral neuropathy were seen and were dose-limiting toxicities [107]. In the ongoing GHSG HD21 trial, BrECADD is being tested as an alternative approach to further reducing chemotherapy-associated acute and long-term toxicity in adults with advanced-stage cHL [105].

BV was also examined in AETHERA, a phase 3 trial, in a post-ASCT setting for high-risk HL [156]. The study revealed a considerable increase in the 5-year PFS rate in the therapy group compared with the placebo group (59% vs. 41%, respectively). Though OS survival did not improve significantly in the BV group compared with the placebo group, fewer patients in the BV group (23 vs. 12) underwent an allo-SCT. Taking into account the likely benefits of BV, the FDA approved the drug in 2015, following this study, for use as a maintenance treatment after ASCT [156]. Additionally, in a retrospective study performed at MD Anderson Cancer Center to examine post-ASCT outcomes, BV exposure prior to ASCT and salvage with vorinostat plus gemcitabine and BuMel was associated with improved results as combined as well as independent factors [130].

At present, BV is being tested in several clinical trials as a replacement for the alkylating medicines used in frontline therapy to prevent secondary cancers. The COG studied the incorporation of BV into the AVE-PC (brentuximab, doxorubicin, bleomycin, vincristine, etoposide, prednisone, and cyclophosphamide) regimen and compared the combination (BV-AVE-PC) against the standard-of-care therapy, ABVE-PC, in a recently finished phase 3 trial, AHOD1331 (NCT02166463) [57,73,121,122]. In the BV-AVE-PC group, the 3-year EFS and OS rates were 92.1% and 99.3%, respectively; in the ABVE-PC group, they were 82.5% and 98.5%, respectively. These study results were recently presented at the 2022 American Society of Clinical Oncology (ASCO) Annual Meeting. This trial essentially showed that BV-AVE-PC ought to replace frontline therapy as the new gold standard for pediatric cHL [121]. 

In a pilot, non-randomized trial (NCT02398240), Hochberg et al. [108] are currently evaluating the effectiveness of upfront BV with combination chemotherapy for the treatment of children, adolescents, and young adults with newly diagnosed HL. A total of four or six cycles of doxorubicin, vinblastine, dacarbazine, BV, and rituximab were given to all patients. The study’s preliminary results are astounding; they show 100% complete response, EFS, and OS rates at a median follow-up of 5 years, as well as a decrease in radiotherapy use, among the intermediate- and high-risk cHL groups. In an effort to establish BV as the first-line therapy for all phases of newly diagnosed cHL, the trial is also investigating BV in newly diagnosed, low-risk cHL patients in light of these findings [108].

On 10 November 2022, the FDA approved BV (in combination with doxorubicin, vincristine, etoposide, prednisone, and cyclophosphamide) for children over the age of 2 years who have newly diagnosed, high-risk cHL. This is BV’s first pediatric approval [157].

### 14.2. Proteasome Inhibitor (Bortezomib)

NF-κB proteins have been identified as potential biomarkers in pediatric HL. Importantly, dysregulation in the NF-κB signaling that leads to NF-κB activation, which is common in lymphoid malignancies, is present in HRS cells. These signaling pathways have been well studied in adults, but limited studies have been carried out in pediatric patients [7]. The COG clinical trial AHOD0031 evaluated patients with intermediate-risk HL to try to understand the role of NF-κB pathway proteins such as Rel-B, NIK, and A20, along with cytoplasmic Rel-A and IKK-β. The researchers concluded that the NF-κB pathway is dysregulated in EBV-positive tumors, and they also noted that those patients with elevated levels of NF-κB pathway proteins had slow responses to therapy [158]. 

Bortezomib is a proteasome inhibitor that has shown promising results for the treatment of pediatric HL. Proteasome inhibitors utilize the mechanism of ubiquitin–proteasome-pathway-dependent degradation of proteins involved in the NF-κB pathway in HRS cells [159]. Thus, bortezomib, which targets the NF-κB pathway and CD30, has great potential as a treatment for HL. Therefore, agents that target the NF-κB pathway and CD30 are promising for therapy. It has been shown that brentuximab vedotin and bortezomib are most effective for treating patients with R/R Hodgkin’s lymphoma [7]. 

A COG phase 2 trial studied bortezomib in combination with isocyanide and vinorelbine in 26 pediatric and young adult patients with R/R HL. They confirmed the overexpression of NF-κB pathway proteins such as Rel-A, Rel-B, NF-κB subunits, and cytoplasmic Rel-B in HL patients. The study showed an improvement in the overall response rate (91%) at the completion of therapy [160]. 

T-cell depletion is commonly seen in patients on bortezomib therapy, and it can make patients susceptible to viral infections such as the reactivation of the varicella-zoster virus [161]. Therefore, antiviral prophylaxis is highly recommended. In addition, bortezomib-associated neuropathy was reported in 37% to 44% of patients in a multiple myeloma clinical trial. It seems that the most important factor in the development of neuropathy is the cumulative dose of bortezomib. Other side effects include gastrointestinal toxicity, neutropenia, and thrombocytopenia, which are common after treatment with several chemotherapy agents.

### 14.3. CD20 Inhibitor (Rituximab)

Rituximab is a monoclonal antibody against the CD20 cell surface marker, which is typically expressed in B-cell non-HL and NLPHL. Rituximab is the standard of care for these diseases, and its use in combination chemotherapy has significantly increased patients’ cure rates [50,162]. It is usually safe, has a tolerable adverse-effect profile, and is not associated with typical cytotoxic toxicities [50].

In cHL, HRS cells express CD20 in 20% to 30% of patients [151]. A heterogeneous inflammatory infiltrate composed of T cells, B cells, natural killer cells, and mast cells surrounds the HRS cells, which account for about 5% of the tumor’s cellularity [108,148]. This TME sends several survival signals that support HRS cells’ survival through the release of a number of cytokines [148,149]. Regulatory B cells make up roughly half of the percentage of the TME’s invading cells, which provided the rationale for targeting CD20+ regulatory B cells and CD20+ HRS cells with rituximab for the treatment of cHL treatment. This rationale has been supported by preclinical data [108,149,151,163].

Rituximab has demonstrated efficacy as a single agent in relapsed cHL and in combination with ABVD (R-ABVD) as a frontline therapy for patients with advanced cHL [164,165,166,167]. R-ABVD and ABVD were compared as upfront therapies for patients with advanced-stage, high-risk cHL in a multicenter, open-label, randomized, phase 2 study (NCT00654732) [168]. The study reported an improvement in the 3-year EFS rate, but it failed to demonstrate any significant difference in the OS rate in the R-ABVD group. Additionally, compared to the patients in the ABVD group, the patients in the R-ABVD group had a higher incidence of neutropenia, anemia, thrombocytopenia, infections, respiratory issues, and neuropathy, which were the most prevalent grade 3–4 toxicities associated with the addition of rituximab [168].

The GHSG trial HD18 explored the benefits of adding rituximab to BEACOPP (R-BEACOPP) with an interim PET/CT-adapted approach to therapy. However, the study did not demonstrate any improvement in patient outcomes [169].

Abuelgasim et al. [170] recently concluded, after a retrospective, real-world data analysis, that the addition of rituximab has no discernible effect on the outcomes of patients with CD20+ cHL and did not recommend rituximab’s use outside of investigational studies. Hence, more clinical trials are warranted to examine the activity of rituximab in combination with biological treatments and/or immunotherapy in patients with cHL.

### 14.4. Immune Checkpoint Inhibitors/PD-L1 Inhibitors

Immune checkpoint inhibitors that target PD-1 show promise for the treatment of HL in the upfront and relapsed settings. The importance of blocking PD-1 lies in its ability to suppress T-cell responses to tumors, leading to ineffective killing [171]. In fact, HL’s overexpression of PD-L1 is thought to be a by-product of chromosome 9p24.1 amplification, a genetic aberration commonly seen in the nodular sclerosing form of HL [172]. Moreover, it has been demonstrated that PD-1 on the surface of T cells is a marker of T-cell exhaustion, especially in T cells that chronically interact with antigens [173]. Thus, the inhibition of PD-1 represents a manner of TME alteration that can lead to a more sensitive tumor and improve killing through cytotoxic T cells. 

In a study examining the use of nivolumab, a PD-1 inhibitor, in patients whose disease relapsed following treatment with ASCT or BV, the results were encouraging; 87% of patients demonstrated a response, and the PFS rate at 24 weeks was 86%. Moreover, only 22% of patients demonstrated a grade 3, drug-related adverse event [174]. Another study by Younes et al. [175] demonstrated similar results; 66% of patients who received nivolumab following the failure of ASCT or BV demonstrated a response [175]. These studies led to the FDA’s approval, in May 2016, of nivolumab for the treatment of R/R HL in patients who did not respond to ASCT and BV [176]. While these studies were performed in trials for adult patients, the results set up the use of nivolumab for the treatment of pediatric R/R HL. In a phase 1/2 trial, nivolumab was found to have a safe toxicity profile for the treatment of R/R HL in children and young adults, as no dose de-escalations or dose-limiting toxicities were observed [177]. 

Another ongoing phase III study by Southwest Oncology Group (SWOG1826) randomized the standard BV-AVD versus nivolumab plus AVD in patients with newly diagnosed advanced-stage(III/IV) cHL of age 12 years and above. This study will explore the possibility of sparing bleomycin-based therapies, and hence the associated toxicity, in front-line treatment regimens for advanced-stage cHL. The results of SWOG1826 are yet pending and eagerly awaited [178]. 

Similarly, pembrolizumab, a PD-1 inhibitor, has demonstrated favorable results when used in the treatment of R/R HL. In the KEYNOTE-204 phase 3 study (NCT02684292), the efficacy of pembrolizumab versus BV was studied in patients 18 years of age or older who had either relapsed following ASCT or were unable to receive an ASCT. The results from the trial were promising, as the patients who received pembrolizumab demonstrated a median PFS duration of 13.2 months compared with 8.3 months for those who received BV, and only 16% of patients who received pembrolizumab experienced serious treatment-related adverse effects [155]. The major adverse events reported in the pembrolizumab group were pneumonitis (4%), neutropenia (2%), and peripheral neuropathy (1%) [155]. The use of pembrolizumab in the treatment of R/R HL in pediatric and adolescent patients who failed at least three lines of therapies was first approved by the Food and Drug Administration (FDA) in May 2017 based on the results of the KEYNOTE-087 trial, which demonstrated that nearly half of the adult patients had a partial response [179]. 

Because of the promising results of the KEYNOTE-204 trial, the approval of pembrolizumab was modified in October 2020 to include pediatric and adolescent patients in whom two lines of therapy had failed [180]. At this time, there are no large-scale, published studies describing the safety and efficacy of pembrolizumab in pediatric and adolescent patients with HL who received the drug for R/R disease. However, Zinzani et al. [181] recently reported that, compared to BV, pembrolizumab demonstrated overall improvements in patient-reported outcomes of health-related quality-of-life measures in the KEYNOTE-204 study. Although the study’s complete results have not been published yet, the findings are compelling enough to make pembrolizumab a preferred choice for R/R cHL patients who are either ineligible for or have experienced a relapse after ASCT [155,181].

A case report of a patient who received pembrolizumab as a maintenance treatment following ASCT demonstrated that the drug has a positive safety profile [182]. Current studies have also indicated that pembrolizumab is safe and efficacious and has promise in pediatric and adolescent patients with HL; however, larger-scale studies are needed in this specific population. 

Another immunotherapy agent that has demonstrated clinical activity in R/R HL is a CD30/CD16A-bispecific antibody, AFM13, which is a first-in-class, tetravalent chimeric antibody (TandAb), manufactured to target CD30+ tumors. AFM13 acts by creating a bridge between CD30+ HRS cells and CD16A+ NK cells as well as macrophages, leading to NK cell activation and tumor cells’ apoptosis [183,184]. AFM13 has previously proven to be efficacious as a monotherapy in R/R HL patients [185]. In addition, the combination of pembrolizumab and TandAb seems to be an encouraging treatment regimen for this patient population [183,184].

## 15. Allo-SCT in R/R cHL

Patients who experience a relapse after ASCT have a grave prognosis, with a median survival rate of 2.4 years [186,187]. The curative potential of allo-SCT can help some of these individuals with R/R cHL. Allo-SCT in HL makes use of the graft-versus-lymphoma effect, as described by many studies, which is analogous to the graft-versus-leukemia effect seen in myeloid leukemias [144,188,189,190].

Although significant toxicities can be seen with allo-SCT, including graft-versus-host disease (GVHD), it is interesting to note that chronic GVHD, in particular, correlates with a decreased probability of relapse, perhaps due to the effects of graft-mediated immune surveillance, which prevents lymphoma recurrence. This finding underscores the beneficial impact of the graft-versus-lymphoma effect [144,191,192]. More prospective studies are needed to explore the benefits and the toxicity profile of allo-SCT patients with R/R cHL [144,193]. Additionally, the ideal timing for and the long-term benefits of allo-SCT, as well as the right patient population for the procedure, are still debated among transplant specialists, with the argument in favor of limiting this approach for patients who relapse after receiving ASCT [144,194,195]. 

The choice of donor and conditioning regimens is also an area of ongoing research. Although myeloablative regimens provide the best opportunity for cure, unfortunately, these regimens also are associated with higher rates of transplant-related mortality [147]. A multivariate analysis revealed that receiving the reduced-intensity conditioning regimen of cyclophosphamide, fludarabine, and low-dose, whole-body irradiation was predictive of better PFS and OS, while positive residual disease at the time of transplant was associated with adverse results [196]. A recent report from the CIBMTR demonstrated that the PFS of patients who relapsed within 12 months of their first transplant was poorer compared with those who relapsed more than 3 years after transplant [197].

Additionally, it is amusing to note that the introduction of post-transplant cyclophosphamide (PT-Cy) as GVHD prophylaxis in haploidentical transplants has shown light on the drug’s additional beneficial effects. PT-Cy not only reduces the incidence of GVHD, but also preserves the graft-versus-lymphoma effect. Hence, haploidentical allo-SCT with PT-Cy has become increasingly popular as a standard approach to treating patients with R/R cHL patients [144,198,199].

In comparison to HLA-matched transplants in cHL, several retrospective investigations have revealed that haploidentical allo-SCT offers a higher PFS as well as relapse-free survival (RFS) [200,201,202]. Interestingly, the overall outcomes seem superior to matched-sibling donor (MSD) transplant but comparable to matched-unrelated donor (MUD) [144,198,200]. Martinez et al. reported RFS of 40% for haploidentical, 28% for MSD, and 38% for MUD; and PFS of 43% for haploidentical, 38% for MSD, and 45% for MUD, respectively [200]. These findings suggest that allo-SCT utilizing a haploidentical donor and PT-Cy is just as feasible as using an HLA-matched donor. It also suggests that allo-SCT is a suitable choice when a conventional donor is not available [144,200]. Prospective trials are warranted to further explore the benefits and feasibility of allo-SCT in this situation [144].

It is established that survival after transplant in HL patients is strongly influenced by the disease’s response to salvage therapy. In the recent CIBMTR report, the 3-year probability (95% CI) of survival among the 8311 patients receiving ASCT for HL between 2009 and 2019 was 89% (88–90%) and 78% (74–81%) for patients with chemosensitive and chemoresistant disease, respectively, while in the 1694 patients who had received allo-SCT for HL between 2009 and 2019, the 3-year probability (95%CI) of survival was 67% (65–70%) and 50% (44–56%) for patients with the chemosensitive and chemoresistant disease, respectively (Figure 2) [143].

## 16. CAR T-Cell Therapy for the Treatment of HL

Anti-CD19 CAR T-cell therapy is a groundbreaking innovation to combat R/R hematological malignancies, most commonly pediatric acute lymphoblastic leukemia [203]. CAR T-cell therapy has not been authorized for use in HL, although multiple clinical trials are underway to determine the safety and efficacy of this cell therapy for the treatment of R/R HL, and the results have been promising. Because nearly all current products such as axicabtagene ciloleucel and tisagenlecleucel target the CD19 marker in acute lymphoblastic leukemia and non-HL, an antigen not expressed in HL, there is a need to generate new CAR T-cell constructs to specifically target the antigens expressed on HL [203,204]. Considering that the RS cells found in HL express CD30, the antigen represents a possible target for CAR T cells. In fact, Ramos et al. [205] conducted two phase 1/2 studies using anti-CD30 CAR T cells for the treatment of R/R HL. They demonstrated an overall response rate of 72%, with 59% of patients demonstrating a complete response, and no patients experiencing neurotoxicity. 

Other studies have evaluated using CAR T cells to alter the TME in order to target HL. In vitro, CAR T cells targeting the CD123 receptor on HRS cells have been found to kill not only tumors but also tumor-associated macrophages, cells that inhibit T-cell stimulation [206]. This finding may demonstrate a vital component of using CAR T-cell therapy for the treatment of HL, considering that a high burden of tumor-associated macrophages has been associated with worse outcomes in patients with HL [207]. The use of anti-CD19 CAR T cells to suppress B cells with an inhibitory influence on cytotoxic T cells has been suggested as a means of indirectly using CAR T cells in the treatment of R/R HL. However, larger studies are needed to reach a conclusion regarding the feasibility of this modality [208]. It is apparent that the treatment of HL using CAR T-cell therapy is currently in its infancy, and much work is needed to identify applicable targets as well as indirect mechanisms that may alter the TME and the cellular milieu in order to influence cytotoxic T cells and their killing of tumor cells.

## 17. Ongoing Research

Several preclinical research studies are ongoing for HL investigating signaling pathways and the TME.

Tumor-associated macrophages (TAMs) and the hyperactivation of the PI3K/AKT pathway impact the pathogenesis of cHL. PI3K is overexpressed in HRS cells as well as the TME. An improved antitumor response was discovered in a pre-clinical model investigating the role of PI3Kδ/γ inhibitor RP6530 at HRS cells and TME. This suggests that PI3Kδ/γ inhibition is a novel therapeutic approach for treating HL patients [209].

Heat-shock proteins (HSPs) are molecular chaperones highly expressed in many malignancies, including leukemias and lymphomas [210]. HSP90 has been found to be overexpressed in cHL cells and seems to play an essential role in their survival due to its cytoprotective effects on cell death and signaling pathways, such as the JAK-STAT pathway [211]. HSP90 inhibition appears to have an impact on cHL cell survival. The possible application of HSP inhibitors in the management of HL is currently being researched. It has been demonstrated that HSP90 inhibitors, such as AUY922, have anti-proliferative effects and make cells more susceptible to apoptosis. While HSP inhibitors have potential as a treatment for HL, more studies are required to establish their effectiveness and safety in this setting [210].

Ongoing clinical studies are focusing on response-based strategies to improve short and long-term outcomes after HL treatment [212].

## 18. Surveillance and Acute and Long-Term Adverse Events Associated with HL Therapy

Long-term survivors of HL are at high risk of various organ toxicities secondary to their therapeutic regimens [213]. In addition to organ damage, survivors of childhood HL have a nearly 20-fold increased risk of developing a second malignancy [213]. The most common secondary malignancies in patients with HL are breast and thyroid cancers, acute myeloid leukemia, and soft-tissue sarcomas [213]. This risk remains elevated even decades after treatment [213]. For this reason, most of these patients require lifelong surveillance and risk-reduction strategies [213]. Fortunately, as therapies have evolved and become more narrowly tailored in pediatric populations, the rates of organ toxicities, late mortality, and secondary malignancies have begun to decline [214]. Table 6 lists the most common long-term sequelae associated with HL therapy.

## 19. Conclusions

HL is a complex, multifactorial malignancy that primarily involves B cells [37]. It accounts for approximately 7% of childhood cancers overall and 1% of childhood cancer deaths in the United States [10]. Most children diagnosed with HL are cured with first-line therapy options and have an excellent prognosis; the 5-year survival rate for early stage disease exceeds 90% [216]. For those individuals in whom initial therapy fails, several novel therapeutic approaches are available, including stem cell transplant, the use of monoclonal antibodies, chimeric antigen receptor T-cell therapy, and immunotherapy [129]. An estimated 40% of patients in whom traditional chemotherapy and radiation treatment regimens fail respond to these second-line therapies [129]. Unfortunately, many survivors of childhood HL must also deal with late long-term effects arising from their therapeutic regimens. This risk remains elevated even decades after treatment [213]. For this reason, most childhood HL survivors require lifelong surveillance and cancer risk reduction strategies [213]. However, as therapies continue to evolve and become more narrowly tailored in pediatric populations, it is likely that the rates of organ toxicities, late mortality, and secondary malignancies will continue to decline [214].

While this review comprehensively covers the historical background, pathophysiology, and recent therapeutic advances in HL, it is important to acknowledge that the field is constantly evolving, and new research findings may emerge after the publication of this article. Additionally, as with any review article, the scope of this paper may limit the depth of coverage on certain aspects of HL, such as specific molecular markers, emerging targeted therapies, or rare refractory disease strains. Further research and specialized studies are always warranted to explore these areas in more detail.

Table 7 summarizes the search strategy utilized by the authors in writing this manuscript.

## Figures and Tables

**Figure 1 ijms-24-10095-f001:**
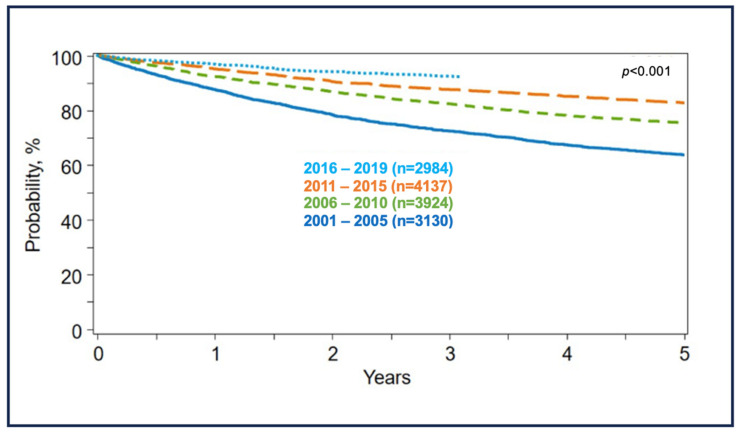
Trends in survival after ASCT for patients with HL in the United States 2001–2019. Footnote: slide taken from CIBMTR US summary slides. (Reference: Auletta, J.J.; Kou, J.; Chen, M.; Shaw, B.E. Current use and outcome of hematopoietic stem cell transplantation: CIBMTR US summary slides, 2021 [143]). Of note, the views expressed in this article are that of the authors and does not reflect the position of the Center for International Blood and Marrow Transplant Research.

**Figure 2 ijms-24-10095-f002:**
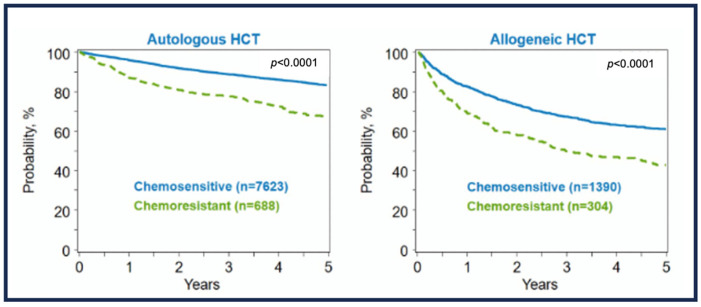
Survival after ASCT or Allo-SCT for HL, in the United States, 2009–2019: slide taken from CIBMTR US summary slides. (Reference: Auletta, J.J.; Kou, J.; Chen, M.; Shaw, B.E. Current use and outcome of hematopoietic stem cell transplantation: CIBMTR US summary slides, 2021 [143]).

**Table 1 ijms-24-10095-t001:** Essential components of the diagnostic workup for HL.

Workup Component	Essential Elements
History	- Detailed history and physical examination for all patients, including: History of B symptoms ^a^ [66]. History of autoimmune disorders, immunodeficiencies, and immunosuppression (i.e., solid organ/stem cell transplant). History of foreign travel. Sexual history (including HIV screening). History of exposure to animals (i.e., cats), as bartonellosis can result in lymphadenopathy. History of exposure to high-risk individuals (i.e., recent inmates, immigrants who may have a history of tuberculosis). Detailed family history (particularly of any other family members with lymphoma) [68].
Physical examination	- Thorough examination of the entire body for lymphadenopathy, with any noted lymphadenopathy described in detail (location, size, tenderness, firmness, and mobility). - Close examination of the liver and spleen for hepatosplenomegaly. - Close examination of the tonsils, base of the tongue, and nasopharynx (Waldeyer ring), as these are nodal tissues [69].
General laboratory tests	- Complete blood count. - Comprehensive metabolic panel. - Erythrocyte sedimentation rate. - C-reactive protein test.
Infectious disease laboratory tests	- HIV testing [70]. - Tuberculosis skin test [71]. - EBV titers [70].
Imaging	- Chest radiograph (anteroposterior and lateral views). - Computed tomography scan of neck, chest, abdomen, and pelvis (with/without contrast). - PET scan.
Procedures	- Bone marrow biopsy and aspiration ^b^. - Lumbar puncture. - Excisional lymph node biopsy ^c^.
Miscellaneous studies	- Genetic/molecular analysis. - Flow cytometry. - FISH. - Tests for the serum cytokines IL 6, IL 10, and soluble CD25 to correlate with systemic symptoms and prognosis [72].

^a^ The presence or absence of systemic B symptoms is a prognostic factor used to determine staging, so ascertaining the presence and timing of these symptoms is critical. ^b^ When a PET scan is available, most centers choose not to perform a bone marrow biopsy because it has a low diagnostic value when compared to a PET scan. ^c^ A core needle biopsy is acceptable if an excisional lymph node biopsy cannot be obtained. Abbreviations: CD, cluster of differentiation; EBV, Epstein–Barr virus; FISH, fluorescence in situ hybridization; HIV, human immunodeficiency virus; IL, interleukin; PET, positron emission tomography.

**Table 2 ijms-24-10095-t002:** The Cotswolds-modified Ann Arbor classification system for HL [8,57].

Stage/Stage Suffix	Description
I	Disease limited to a single lymph node region or one type of extra-lymphatic lymphoid tissue (e.g., spleen, thymus, Waldeyer ring)
II	Disease involving two or more nodal groups on the same side of the diaphragm
III	Involvement of two or more lymph node regions or extra-lymphatic organs on both sides of the diaphragm
	III1: With or without splenic, hilar, celiac, or portal nodes
	III2: With para-aortic, iliac, or mesenteric nodes
IV	Diffuse or disseminated disease (affecting one or more extra-lymphatic organs, e.g., lung, bone, liver) with or without nodal involvement
A	Absence of B symptoms
B	Presence of B symptoms: fever, night sweats, and/or ≥10% weight loss within the preceding 6 months
X	Bulky disease (i.e., nodal mass greater than one-third of the intrathoracic diameter on a chest X-ray) For pediatric HL: bulky disease is also defined as extra-mediastinal mass ≥6 cm in diameter European Network: ≥200 mL of contiguous tumor volume
E	Extra-nodal extension on one side of the diaphragm by limited direct extension

**Table 3 ijms-24-10095-t003:** The Lugano classification system for HL *^,§^ [79].

Stage I—Involvement of a single lymph node region (e.g., cervical, axillary, inguinal, or mediastinal) or lymphoid structure such as the spleen, thymus, or Waldeyer’s ring.
Stage II—Involvement of two or more lymph node regions or lymph node structures on the same side of the diaphragm.
Stage III—Involvement of lymph node regions or lymphoid structures on both sides of the diaphragm.
Stage IV—Diffuse or disseminated involvement of one or more extra-nodal organs ** or tissue with or without associated lymph node involvement.

* All stages are subclassified to indicate the absence (A) or presence (B) of the systemic symptoms of significant unexplained fever, night sweats, or unexplained weight loss exceeding 10% of body weight during the 6 months prior to diagnosis. ^§^ Bulky disease: A single nodal mass, in contrast to multiple smaller nodes, of 10 cm or ≥one-third of the transthoracic diameter at any level of thoracic vertebrae as determined by CT; the longest measurement recorded by CT scan. The term “X” (used in the Ann Arbor staging system) is no longer necessary. ** The designation “E” refers to extra-nodal contiguous extension (i.e., proximal or contiguous extra-nodal disease). Abbreviations: CT, computerized tomography.

**Table 4 ijms-24-10095-t004:** The EuroNet-PHL treatment groups for pediatric cHL.

Treatment Group (TG)	Ann Arbor Stage
TG-1	Stages IA/B and IIA
TG-2	Stages IEA/B, IIEA, IIB, and IIIA
TG-3	IIEB, IIIEA/B, IIIB, and IVA/B

**Table 5 ijms-24-10095-t005:** HL risk stratification and treatment regimens.

Risk Level	Definition	Chemotherapy Regimen
Low risk	Ann Arbor stages IA and IIA Non-bulky disease No B signs/symptoms	Four cycles of VAMP plus LD-IFRT for those who achieve a complete response [110].
		Four cycles of COPP/ABV plus LD-IFRT [111,124].
		High-dose ABVE administered for two to four courses (depending on response), plus LD-IFRT [109].
		For males: OEPA; for females: OPPA. All patients follow with LD-IFRT [114,125].
Intermediate risk	Ann Arbor stages IB and IIB (both without bulk) OR Stages IA and IIA (both with bulk) OR Stages IIAE and IIIA (regardless of bulk)	Six cycles of COPP/ABV plus LD-IFRT [124,126,127].
		ABVE-PC administered for three to five courses, depending upon response; followed by LD-IFRT [75,119].
		Two cycles of OEPA (for males) or OPPA (for females), followed by two cycles of COPP (for females) or COPDAC (for males), plus LD-IFRT [75,114,125,128].
High risk	Ann Arbor stages IIIB and IV	ABVE-PC, administered for three to five courses (depending upon response), followed by LD-IFRT [75,119].
		Two cycles of OEPA (for males) or OPPA (for females), followed by two cycles of COPP (for females) or COPDAC (for males) plus LD-IFRT [75,114,128].
		Two cycles of cytarabine/etoposide, COPP/ABV, and CHOP plus LD-IFRT [126,127].
		Four cycles of BEACOPP with subsequent therapy dependent upon response. Rapid responders: four cycles of COPP/ABV without IFRT (for females) or two cycles of ABVD with IFRT (for males). Slow responders: four additional cycles of BEACOPP plus IFRT [118].

Abbreviations: VAMP, vinblastine, doxorubicin, methotrexate, and prednisone; LD-IFRT, low-dose involved field radiotherapy; COPP, cyclophosphamide, vincristine, procarbazine, and prednisone; ABV, doxorubicin, bleomycin, and vinblastine; ABVE, doxorubicin, bleomycin, vincristine, and etoposide; OEPA, vincristine, etoposide, prednisone, and doxorubicin; OPPA, vincristine, procarbazine, prednisone, and doxorubicin; ABVE-PC, doxorubicin, bleomycin, vincristine, and etoposide, prednisone, cyclophosphamide (ABVE with prednisone and cyclophosphamide); COPDAC, cyclophosphamide, vincristine, prednisone, and dacarbazine; CHOP, cyclophosphamide, doxorubicin, vincristine, and prednisone; BEACOPP, bleomycin, etoposide, doxorubicin, cyclophosphamide, vincristine, procarbazine, and prednisone; ABVD, doxorubicin, bleomycin, vinblastine, and dacarbazine.

**Table 6 ijms-24-10095-t006:** Long-term adverse events secondary to HL therapy [213,215].

Affected Area	Adverse Effects
Growth and development	- Growth stunting secondary to radiation * exposure. - Long-term adverse effects on bone and soft tissue growth in pediatric patients.
Endocrine system	- Hypothyroidism. - Need for thyroid hormone replacement therapy ** (20–30% of HL survivors).
Reproductive system	- Gonadal dysfunction ^$^ (common in both male and female survivors of HL).
Pulmonary system	- Irradiation pneumonitis. - Interstitial pneumonia. - Pulmonary fibrosis. - Decreased pulmonary function.
Cardiovascular system	- Cardiotoxicity ^¶^ (cardiomyopathy, valvular injury, cardiac conduction defects). - Pericarditis. - Pulmonary fibrosis. - Accelerated atherosclerosis. - Increased risk of stroke.
Secondary malignancies ^#^	- Breast and thyroid cancers. - Acute myeloid leukemia. - Soft-tissue sarcomas.

* Adverse radiation effects are dose-dependent. ** Hypothyroidism typically occurs approximately 1–10 years after radiation and is the most common adverse event. ^$^ The risk of gonadal toxicity is increased in patients who receive alkylating agents (i.e., cyclophosphamide, ifosfamide). ^¶^ Risk of cardiotoxicity increases in patients irradiated before age 20. ^#^ There is a 20-fold increased risk of developing a second malignancy in HL survivors.

**Table 7 ijms-24-10095-t007:** Search Strategy.

Items	Specification
Date of search	1 September 2022 to 10 May 2023
Databases, search engines, and other resources utilized	PubMed, Google Scholar, clinicaltrials.gov (accessed on 9 May 2023), NCCN guidelines, American Society of Hematology conference presentations, American Society of Hematology conference presentations
Search terms used	Hodgkin lymphoma, new drugs, brentuximab, checkpoint inhibitors, chimeric antigen T cells, targeted therapy
Type of studies used	Review articles, systematic reviews, clinical trials, conference abstracts, basic science articles
Inclusion and exclusion criteria	Only English-language studies were used
Selection process	Studies were selected independently by the authors writing their respective sections; consensus was obtained by multiple revisions among the authors
Total articles	Over 215 articles are cited in total

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
