# Peer review of "Classical Hodgkin Lymphoma: From Past to Future—A Comprehensive Review of Pathophysiology and Therapeutic Advances"

_ijms, 2023, doi:10.3390/ijms241210095_

Round 1

Reviewer 1 Report

The authors have provided a detailed review on Hodgkin lymphoma. This is an interesting, well-organized review, however, there are several issues that should be addressed:

1. The manuscript lacks scientific/ biological insight of HL pathogenesis. Most information in the review is a clinical HL rather than scientific.  Authors should add scientific/ biology insight into the pathogenesis of HL.

2. There are no illustrations in the manuscript. Authors should add illustrations to make the review reader-friendly

3.  There is no information in the manuscript on new strategies that are under investigation by different research groups. For example, the role of heat shock proteins and their inhibitors for the HL treatment. Authors should do a comprehensive literature review on new strategies that are currently under investigation by researchers in the field of HL

4. Authors should describe the difference between Hodgkin cells and Reed-Sternberg cells. 

Minor comments:

Nivolumab is a PD-1 inhibitor, not PD-L1

The following sentence should be rephrased: "Proteasome inhibitors use the ubiquitin-proteasome pathway as a mechanism of protein degradation in cells such as NF-κB pathway proteins, such as HRS cells [161]"

Author Response

Dear Drs. El-Gamal and D’Angelo, 

We sincerely appreciate the opportunity to revise our manuscript " Classical Hodgkin Lymphoma: From Past to Future – A Comprehensive Review of Pathophysiology and Therapeutic Advances " ijms-2399454. We want to thank the reviewers for the time and effort they took in reviewing our manuscript and the constructive feedback they provided, which we have used to revise the manuscript we now submit. 

To assist your evaluation of this manuscript, we have addressed the reviewers' comments in our point-by-point response below and have made the corresponding changes to the revised manuscript. The reviewer's comments are highlighted in bold, and our answers can be found underneath. Where appropriate, we have included verbatim text from our manuscript to show the changes made in response to the reviewer's suggestions. 

We would be very obliged for your time and attention in reviewing our revised manuscript.

Thank you again for your grace.

Respectfully,

Faryal Munir, MD

Fellow, Pediatric Hematology/Oncology

MD Anderson Cancer Center

The University of Texas, Division of Pediatrics

FMunir@mdanderson.org
Phone (work): 713-540-8013

Reviewer 1 Comments:

The authors have provided a detailed review on Hodgkin lymphoma. This is an interesting, well-organized review, however, there are several issues that should be addressed:

  1. The manuscript lacks scientific/ biological insight of HL pathogenesis. Most information in the review is a clinical HL rather than scientific. Authors should add scientific/ biology insight into the pathogenesis of HL.

Reply 1: Thank you for the observation of this discrepancy.

Changes in the text: Molecular biology section is incorporated into the pathophysiology section and additional references are quoted.

  1. There are no illustrations in the manuscript. Authors should add illustrations to make the review reader-friendly

Reply 2: Thank you for the great recommendation.

Changes in the text: Illustrations are added to make the manuscript more reader-friendly.

  1. There is no information in the manuscript on new strategies that are under investigation by different research groups. For example, the role of heat shock proteins and their inhibitors for the HL treatment. Authors should do a comprehensive literature review on new strategies that are currently under investigation by researchers in the field of HL.

Reply 3: Thank you for bringing this to our attention.

Changes in the text: A section on ongoing research on HL is added, and studies are cited accordingly.

  1. Authors should describe the difference between Hodgkin cells and Reed-Sternberg cells.

Reply 4: Thank you for raising this point.

Changes in the text: Additional description highlighting the differences of Hodgkin and RS cells is added in the pathophysiology section.

  1. Nivolumab is a PD-1 inhibitor, not PD-L1

Reply 5: Thank you for recognizing this error.

Changes in the text: Corrected at all respective places.

  1. The following sentence should be rephrased: "Proteasome inhibitors use the ubiquitin-proteasome pathway as a mechanism of protein degradation in cells such as NF-κB pathway proteins, such as HRS cells [161]"

Reply 6: Thank you for identifying this.

Changes in the text: Statement is revised.

Reviewer 2 Report

1. Check the abbreviations throughout the manuscript and introduce the abbreviation when the full word appears the first time in the abstract and the remaining for the text and then use only the abbreviation (For example, Hodgkin lymphoma (HL), Classical Hodgkin lymphoma (cHL), etc.,). Make a word abbreviated in the article that is repeated at least three times in the text, not all words to be abbreviated.

2. How many articles obtained from each of the search engines? What is the inclusion and exclusion criteria? What is the type of article included in this manuscript? How many articles are included for this manuscript? A diagram depicted the literature search should be included (If possible only) for better understanding and outcome.

3. The table legends should be improved and a proper footnote should be given. All legends should have enough description for a reader to understand the table without having to refer back to the main text of the manuscript. For example, the necessary expansion may be given for abbreviations used.

4. The limitation of the present investigation may be given along with conclusion or under separate heading for understanding the concepts clearly.

1. The English need improvement since there are some grammatical and syntax errors in the manuscript. For example, in line number 56, the word “disorders” may be as “are disorders”;

·         in line number 153, “development” as “the development”;

·         in line number 203, “a diffuse” as “diffuse”;

·         in line number 279, “for [69]” as “[69]”;

·         in line number 377, “biopsy” as “a biopsy”;

·         in line number 427, “lead” as “led”;

·         in line number 510, “the patients” as “patients”;

·         in line number 570, “and to” as “and”;

·         in line number 595, “a sluggish” as “sluggish”;

·         in line number 598, “use” as “the use”;

·         in line number 690, “a shorter” as “shorter”;

·         in line number 732, “transplant” as “the transplant”;

·         in line number 773, “post-ASCT” as “a post-ASCT”;

·         in line number 787, “BV” as “of BV”;

·         in line number 800, “was given” as “were given”;

·         in line number 812, “has been” as “have been”;

·         in line number 826, “CD30 is” as “CD30 are”;

·         in line number 847, “chemotherapy” as “with chemotherapy”;

·         in line number 917, “adult” as “the adult”;

·         in line number 986, “the use” as “use”;

·         in line number 1004, “inhibitory” as “an inhibitory”.

The grammar mistakes which are not mentioned here are also to be checked and corrected properly.

2. There are some typing mistakes as well, and authors are advised to carefully proof-read the text. For example,

·         in line number 246, the words “large cell” may be as “large-cell”;

·         in line number 297, “IIB/IIIA ,” as “IIB/IIIA,”;

·         in line number 387, “early stage” as “early-stage”;

·         in line number 432, “deescalation” as “de-escalation”;

·         in line number 558, “long term” as “long-term”;

·         in line number 597, “5-years” as “5 years”;

·         in line number 611, “adapted” as “adopted”;

·         in line number 687, “in for” as “for”;

·         in line number 826, “Therefore ,” as “Therefore,”;

·         in line number 832, “Rel-A ,” as “Rel-A,”;

·         in line number 837, “varicella zoster” as “varicella-zoster”;

·         in line number 853, “accounts” as “account”;

·         in line number 931, “in a” as “of a”;

·         in line number 977, “at 38%” as “38%”;

·         in line number 1000, “but have” as “but”.

The typos not mentioned here are also to be checked and corrected properly. 

Author Response

Reviewer 2:

Comments and Suggestions for Authors:

  1. Check the abbreviations throughout the manuscript and introduce the abbreviation when the full word appears the first time in the abstract and the remaining for the text and then use only the abbreviation (For example, Hodgkin lymphoma (HL), Classical Hodgkin lymphoma (cHL), etc.,). Make a word abbreviated in the article that is repeated at least three times in the text, not all words to be abbreviated.

Reply 1: Thank you for highlighting this.

Changes in the text: All the abbreviations are re-checked and corrected where necessary to the best of our efforts. Journal editors are welcome to convert text into abbreviations or suggest where they deem appropriate.

  1. How many articles obtained from each of the search engines? What is the inclusion and exclusion criteria? What is the type of article included in this manuscript? How many articles are included for this manuscript? A diagram depicted the literature search should be included (If possible only) for better understanding and outcome.

Reply 2: Thank you for the great recommendation.

Changes in the text:  A search strategy table is provided with the requested details.

  1. The table legends should be improved and a proper footnote should be given. All legends should have enough description for a reader to understand the table without having to refer back to the main text of the manuscript. For example, the necessary expansion may be given for abbreviations used.

Reply 3: Thank you for bringing this to my attention.

Changes in the text: All tables are revised with improved legends and abbreviations for the convenience of the reader.

  1. The limitation of the present investigation may be given along with conclusion or under separate heading for understanding the concepts clearly.

Reply 4: Thank you for the suggestion.

Changes in the text: Statement reflecting the limitations of our review article are added in the conclusion section.

  1. Comments on the Quality of English Language:

Reply 5: Thank you for the suggestions.

Changes in the text: Please see the answer in front of individual suggestions. Please note that due to the addition of few more text paragraphs, the new line numbers have changed. However, the suggested corrections are appropriately made.

  1. The English need improvement since there are some grammatical and syntax errors in the manuscript. For example, in line number 56, the word “disorders” may be as “are disorders”; N/A
  • in line number 153, “development” as “the development”; Corrected
  • in line number 203, “a diffuse” as “diffuse”; N/A
  • in line number 279, “for [69]” as “[69]”; Corrected, added the missing word “HL”.
  • in line number 377, “biopsy” as “a biopsy”; N/A
  • in line number 427, “lead” as “led”; Corrected.
  • in line number 510, “the patients” as “patients”; Corrected.
  • in line number 570, “and to” as “and”; Corrected, added the missing word “exposure”
  • in line number 595, “a sluggish” as “sluggish”; N/A
  • in line number 598, “use” as “the use”; Corrected
  • in line number 690, “a shorter” as “shorter”; N/A
  • in line number 732, “transplant” as “the transplant”; Corrected
  • in line number 773, “post-ASCT” as “a post-ASCT”; N/A
  • in line number 787, “BV” as “of BV”; Corrected
  • in line number 800, “was given” as “were given”; Corrected.
  • in line number 812, “has been” as “have been”; Corrected.
  • in line number 826, “CD30 is” as “CD30 are”; Corrected.
  • in line number 847, “chemotherapy” as “with chemotherapy”; N/A
  • in line number 917, “adult” as “the adult”; Corrected.
  • in line number 986, “the use” as “use”; Corrected.
  • in line number 1004, “inhibitory” as “an inhibitory”. Corrected.

The grammar mistakes which are not mentioned here are also to be checked and corrected properly.

  1. There are some typing mistakes as well, and authors are advised to carefully proof-read the text. For example,
  • in line number 246, the words “large cell” may be as “large-cell”; Corrected.
  • in line number 297, “IIB/IIIA ,” as “IIB/IIIA,”; Corrected.
  • in line number 387, “early stage” as “early-stage”; Corrected.
  • in line number 432, “deescalation” as “de-escalation”; Corrected.
  • in line number 558, “long term” as “long-term”; Corrected.
  • in line number 597, “5-years” as “5 years”; Corrected.
  • in line number 611, “adapted” as “adopted”; Corrected.
  • in line number 687, “in for” as “for”; Corrected.
  • in line number 826, “Therefore ,” as “Therefore,”; Corrected.
  • in line number 832, “Rel-A ,” as “Rel-A,”; Corrected.
  • in line number 837, “varicella zoster” as “varicella-zoster”; Corrected.
  • in line number 853, “accounts” as “account”; Corrected.
  • in line number 931, “in a” as “of a”; Corrected.
  • in line number 977, “at 38%” as “38%”; Corrected.
  • in line number 1000, “but have” as “but”. Corrected.

Round 2

Reviewer 1 Report

The authors have addressed all the comments 

Author Response

Thank you very much. We would like to express our utmost gratitude for your gracious acknowledgment and review of our manuscript. Your invaluable input is deeply appreciated by our team.

Sincerely,

Faryal Munir, MD

Reviewer 2 Report

1. There are some grammatical, alignment and typographical errors are noted in the manuscript and it should be thoroughly checked and corrected throughout the manuscript. For example,

·         in line number 54, the word “Hodgkin” may be as “Hodgkin's”;

·         in line number 168, “(cHL),and” as “(cHL), and”;

·         in line number 235, “a diffuse” as “diffuse”;

·         in line number 388,  “stage I or II disease” as “stage I or II diseases”;

·         in line number 716, “disease” as “the disease”;

·         in line number 815, “post-ASCT” as “a post-ASCT”.

2. The suggestions are not carried out properly. Check the abbreviations throughout the manuscript and introduce the abbreviation when the full word appears the first time in the abstract and the remaining for the text and then use only the abbreviation. For example, in line numbers 58 and 66, the authors used full form “Hodgkin lymphoma” instead of short form “HL”. These types corrections need to be checked properly for all other abbreviations used in the manuscript.

3. The technical terms (Latin Phrase) “in vitro (in line number 1075)” should be italic and it should be checked all over the manuscript. 

Author Response

Reviewer 2 Comments and Suggestions for Authors:

  1. There are some grammatical, alignment and typographical errors are noted in the manuscript and it should be thoroughly checked and corrected throughout the manuscript. For example…

Reply 1: Thank you for bringing our attention to these areas needing improvement and grammatical correction. Please see the answer in front of individual suggestions.

  • in line number 54, the word “Hodgkin” may be as “Hodgkin's”; Corrected.
  • in line number 168, “(cHL),and” as “(cHL), and”; Corrected.
  • in line number 235, “a diffuse” as “diffuse”; Corrected.
  • in line number 388,  “stage I or II disease” as “stage I or II diseases”; N/A
  • in line number 716, “disease” as “the disease”; Corrected.
  • in line number 815, “post-ASCT” as “a post-ASCT”. Corrected.

  1. The suggestions are not carried out properly. Check the abbreviations throughout the manuscript and introduce the abbreviation when the full word appears the first time in the abstract and the remaining for the text and then use only the abbreviation. For example, in line numbers 58 and 66, the authors used full form “Hodgkin lymphoma” instead of short form “HL”. These types of corrections need to be checked properly for all other abbreviations used in the manuscript.

Reply 2: Thank you for pointing this out.

Changes in the text: All the abbreviations for “HL” are re-checked and corrected where necessary to the best of our efforts. Additionally revised the terms for “cHL”, “stem cell transplantation”, “SCT” “ASCT”, “allo-SCT” “relapse or refractory”, “R/R” etc. which are modified at several places for the sake of consistency. Nonetheless, journal editors are also welcome to convert text into abbreviations or suggest where they deem appropriate.

  1. The technical terms (Latin Phrase) “in vitro (in line number 1075)” should be italic and it should be checked all over the manuscript.

Reply 3: Thank you for highlighting this.

Changes in the text: The Latin phrases are changed to be in Italics.
